# Low-Waste Synthesis and Properties of Highly Dispersed NiO·Al₂O₃ Mixed Oxides Based on the Products of Centrifugal Thermal Activation of Gibbsite

Aleksey V. Zhuzhgov [1,*], Lyubov A. Isupova [1], Evgeny A. Suprun [1] and Aleksandr S. Gorkusha [1,2]

1    Boreskov Institute of Catalysis, pr. Lavrentieva, 5, 630090 Novosibirsk, Russia; isupova@catalysis.ru (L.A.I.); suprun@catalysis.ru (E.A.S.); deepforesttt922@gmail.com (A.S.G.)
2    Department of Physics, Novosibirsk State University, Street Pirogova, 1, 630090 Novosibirsk, Russia
*    Correspondence: zhuzhgov@catalysis.ru

**Abstract:** This study revealed an increased reactivity of centrifugally thermoactivated products of gibbsite toward aqueous solutions of nickel nitrate at room temperature as well as under hydrothermal conditions. X-ray, thermal, microscopy, adsorption and chemical analysis methods were used to investigate and demonstrate the possibility of obtaining highly loaded mixed aluminum–nickel oxide systems, with a nickel content of ca. 33 wt.%, using a hydrochemical treatment at room temperature or a hydrothermal treatment of suspensions of the product of the centrifugal thermal activation of gibbsite in aqueous solutions of nickel nitrate. It was shown that the thermal treatment of xerogels—hydrochemical interaction products—in the range of 350–850 °C led to the formation of NiO phases and highly dispersed solid solutions of nickel based on the NiAl₂O₄ spinel structure, with different ratios and a high specific surface area of 140–200 m²/g. A hydrochemical treatment of suspensions at room temperature ensures that the predominant formation of the NiO phase is distributed over the surface of the alumina matrix after calcination, whereas hydrothermal treatment at 150 °C leads to a deeper interaction of the suspension components at the treatment step, which occurs after the thermal treatment of the formed xerogel in the predominant formation of poorly crystallized NiAl₂O₄ spinel ("protospinel"). The considered method makes it possible to obtain complex aluminum–nickel oxide systems with different phase ratios, decreases the number of initial reagents and synthesis steps, completely excludes waste and diminishes the total amount of nitrates by 75 wt.% compared to the classical nitrate scheme for the coprecipitation of compounds with a similar elemental composition.

**Keywords:** gibbsite; product of centrifugal thermal activation of gibbsite; mixed oxides NiO·Al₂O₃; protospinel

## 1. Introduction

Aluminum–nickel oxide catalytic systems, specifically those having the structure of stoichiometric spinels (NiAl₂O₄, where Ni constitutes 33 wt.%), are of great practical interest in the application of the dehydrogenation/hydrogenation of hydrocarbons. For example, highly loaded aluminum–nickel systems, particularly with a stoichiometric composition, as in NiAl₂O₄, are employed as catalysts for the hydrogenation of furfural [1–3] or acetone [4,5], the conversion of methane, ethanol and glycerol into valuable hydrocarbons and hydrogen [6–8] and for CO₂ methanation [9]. Aluminum–nickel compositions are promising for the development of improved, new-generation catalysts for the cryogenic catalytic process of hydrogen ortho–para conversion [10–12] and fine catalytic purification of process gases. In particular, the removal of oxygen impurities from hydrogen [13], which is an important aspect of hydrogen generation processes, purifies for liquefaction and provides long-term storage and long-distance transportation [10,12]. It is known that hydrogen in the liquid aggregate state (boiling at the normal pressure and cryogenic temperature of 21–23 K) is the most promising for its transportation and storage abilities in large amounts.

This has been implemented or is being implemented by the Japanese companies Kawasaki Heavy Industries, HESC, NEDO, HySTRA, IWATANI and ENEOS [14,15]. In the overall processing chain, starting from the hydrogen production, the fine catalytic purification to remove microimpurities, up to the transformation into the liquid cryogenic state via the joint ortho–para conversion of hydrogen isomers, is an essential component and is necessary to use catalytic systems, among which the Ni-aluminum compositions may be quite marketable.

Various methods employed for the synthesis of such materials, their advantages and drawbacks are reported in the literature. Aluminum–nickel and other single- and multicomponent materials containing transition and nontransition metals are commonly synthesized by the coprecipitation method (the sol–gel process) [16–19]. The drawbacks of this method include the necessity to use a large number of initial reagents (salts, acids and alkalis), the formation of wastes due to the removal of impurities by washing the precipitates and the formation of effluent gases upon drying and thermal treatment, which should be utilized.

A commonly used method is the incipient wetness impregnation of the preliminarily prepared supports based on aluminum oxides or their precursors, hydroxides, with solutions of various metal salts, particularly nickel [6]. Incipient wetness impregnation has limitations on the concentration of active components introduced into the pores of support, which are related to the moisture absorption limit of the support and the solubility of salts in the impregnating aqueous solutions.

Mechanochemical methods for the treatment of aluminum–nickel systems were reported [2,20–22]. For example, the "aging" of mechanically activated gibbsite in aqueous solutions of $M^{2+}(NO_3)_2$ nitrates (where $M^{2+}$ = Zn, Cu, Ni, Co) was comprehensively studied at room temperature and under autoclave treatment [23,24]. It was shown that the "aging" of the initial (nonactivated) gibbsite at room temperature did not provide the formation of complex aluminum compounds with the corresponding cation. The formation of layered double hydroxides (LDH) was observed only in the case of mechanochemically activated gibbsite products under hydrothermal treatment at 150 °C for 48 h. Such methods included a preliminary mechanical activation of the initial individual hydroxides, oxides and salts of metals or their mixtures, with subsequent hydrochemical treatment in aqueous media (or without it) and drying and thermal treatment, which ensures a decrease in the formation temperature of the final products and allows for the dispersed materials with the specified properties in comparison with the conventional high-temperature sintering of such mixtures to be obtained, which requires much higher temperatures of the synthesis. However, the indicated method did not have wide application due to the lack of necessary industrial equipment.

Along with the mechanochemical activation of gibbsite powder, the most widely known method employed for the production of supports, catalysts and desiccants includes the fast heating of powdered gibbsite in various devices, the so-called "flash" processes [25–27]. Of note, there are two methods developed at the Boreskov Institute of Catalysis: thermochemical activation (TCA) [27] and centrifugal thermal activation (CTA) [26]. Thermoactivation products of gibbsite (CTA-G and TCA-G), in comparison with the initial gibbsite, possess an increased chemical activity toward electrolytes—higher solubility in acids and alkalis [26,27]. Owing to these features, the products obtained by the fast heating of powdered gibbsite, specifically by the CTA method, can be promising for the formation of complex aluminum–nickel systems with different $NiO/Al_2O_3$ ratios, particularly with the $NiAl_2O_4$ spinel structure. Thus, an increased reactivity of CTA-G products can be expected upon contact with solutions of nickel salts, which will ensure the formation of Ni-aluminum materials with the specified elemental composition under mild conditions without classical steps of coprecipitation (the sol–gel process), high-temperature sintering and mechanochemical activation.

Our studies [28–30] have demonstrated the possibility of using the product of the centrifugal thermal activation of gibbsite (CTA-G) to obtain Mg- and Ba-aluminum systems

with the stoichiometric composition $MeAl_2O_4$ (where Me = $Mg^{2+}$, $Ba^{2+}$) by hydrochemical treatment of the CTA-G thermoactivation products in solutions of magnesium or barium nitrates under hydrothermal or "mild" (room temperature) conditions. In comparison with the conventional ceramic method, this allows for a decrease in the calcination temperature, obtaining a highly dispersed product (magnesium or barium aluminate). In comparison with the classical coprecipitation method, such hydrochemical treatment considerably decreases the number of initial reagents and the number of technological steps and minimizes or completely excludes the formation of wastewater and harmful nitrogen oxides ($NO_x$) in off-gases during thermal treatment in the case of partial washing by the precipitation of the occluded admixtures. This makes the proposed method quite attractive for use in various applications, particularly in the preparation of supports and catalysts with a low surface acidity, which is important in the synthesis of catalytic systems. This is essential for the dehydrogenation/hydrogenation of hydrocarbons as well as for the development and improvement of materials for hydrogen energetics.

The goal of this study was to synthesize highly loaded aluminum–nickel oxide systems by the interaction of the centrifugal thermoactivation product of gibbsite in nickel nitrate solutions upon hydrochemical treatment at room temperature and hydrothermal treatment at 150 °C.

## 2. Materials and Methods

The product of the centrifugal thermal activation of gibbsite (CTA-G) IC-02-76 (specs 2175-040-03533913-2007) was obtained using gibbsite (G) GD 000 (specs 1711-99-039-2000) purchased from the Achinsk Alumina Plant as a raw material. The content of admixtures in the initial gibbsite was as follows (wt.%): Fe = 0.002, Na = 0.11, K = 0.033 and Si = 0.014. The specific surface area of the initial gibbsite powder did not exceed 1 $m^2$/g. Losses upon calcination at 850 °C constituted 34 wt.%. The mean particle size of the initial gibbsite powder was 100 μm (before thermal activation and milling). Nickel nitrate hexahydrate $Ni(NO_3)_2 \cdot 6H_2O$ (analytically pure, VECTON Inc., Tokyo, Japan) served as the initial Ni-containing feedstock.

The thermoactivation of gibbsite was carried out in a centrifugal drum-type flash reactor (Tseflar$^{TM}$) at a temperature of 540 ± 5 °C in electric heaters, a drum rotary speed of 60 rpm and the consumption of the initial powder was 40 kg/h. The CTA-G product was then milled in a ball mill for 6 h to obtain a powder with a mean particle size of ca. 15 μm. The weight loss upon calcination of the obtained CTA-G product at 850 °C (4 h) was ca. 12–13 wt.%.

To synthesize the aluminum–nickel samples, CTA-G was loaded in the preliminarily prepared nickel nitrate solution so that the ratio of cations corresponded to the stoichiometric nickel aluminate ($NiAl_2O_4$). The suspension with the initial pH value of ca. 4.5 was subjected to hydrothermal treatment at 150 °C for 4 h under stirring at 120–150 rpm or held under "mild" conditions at room temperature and under hydrothermal conditions at 150 °C for 4 h. The gels that formed as a result of hydrochemical treatment without preliminary washing were dried at 110 °C for 6 h to obtain a xerogel. The subsequent thermal treatment was performed in a muffle furnace at 350–850 °C in air for 4 h. Samples in the text are denoted as Ni-Al($T_1$)-$T_2$, where $T_1$ is the temperature of hydrochemical treatment (hydration) of suspensions (°C) and $T_2$ is the final treatment temperature of the sample (°C). Designations of samples used in this study are described below in Table 1.

The chemical composition of the prepared samples was determined by inductively coupled plasma atomic emission spectroscopy on an OP-TIMA 4300 DV instrument (PERKIN ELMER).

The phase composition of the samples was revealed by X-ray diffraction (XRD) analysis on an ARL-X'TRA diffractometer using CuKα radiation (λ = 1.5418 Å). The samples were scanned point-by-point with 0.05° increments in a 2θ range of 5–70°.

**Table 1.** Designation and description of samples.

| Sample Designation | Description |
| --- | --- |
| G | initial crystal gibbsite (aluminum hydroxide $\gamma$-Al(OH$_3$)) |
| CTA-G | product of centrifugal thermal activation of crystal gibbsite |
| NiAl(25)-110 | product of room temperature hydration (interaction) of CTA-G with aqueous solution of Ni$^{2+}$ nitrate and subsequent drying at 110 °C |
| NiAl(150)-110 | product of hydration (interaction) of activated CTA-G with solution of Ni$^{2+}$ nitrate under hydrothermal treatment at 150 °C and subsequent drying at 110 °C |
| NiAl(25)-550 | product of NiAl(25)-110 thermal treatment at 550 °C |
| NiAl(25)-850 | product of NiAl(25)-110 thermal treatment at 850 °C |
| NiAl(150)-350 | product of NiAl(150)-110 thermal treatment at 350 °C |
| NiAl(150)-550 | product of NiAl(150)-110 thermal treatment at 550 °C |
| NiAl(150)-850 | product of NiAl(150)-110 thermal treatment at 850 °C |

Thermal analysis of the samples was carried out using an STA 449C Jupiter synchronous thermal analysis instrument (NETZSCH). Corundum crucibles were used to study the samples. The rate of air supply to the sample chamber was 30 mL/min; an inert gas (argon) was supplied to the weighing block at a rate of 20 mL/min. The samples were heated at a rate of 2 °C/min from room temperature to 50 °C and held at this temperature for 30 min. Next, the temperature-programmed heating to 850 °C was performed at a rate of 10 °C/min.

The textural characteristics were estimated by the low-temperature nitrogen desorption at 77 K on an automated Quadrasorb-EVO Quantachrome device (USA). The samples were examined in the form of powders with their preliminary evacuation at 300 °C for 2 h. Methods for measuring and calculating the texture parameters complied with ASTM D3663, ASTM D4820, ASTM D1993 and UOP425-86 standards.

A temperature-programmed reduction (H$_2$-TPR) of the samples was carried out on a Chemosorb device (OOO Neosib, Novosibirsk, Russia). Before the reduction, samples (50 mg, fraction 0.25–0.5) were heated for 1 h in an Ar atmosphere at 200 °C. The reduction was carried out at a heating rate of 10 °C/min in a 9.7 vol.% H$_2$/Ar flow. The absorption of H$_2$ was determined with a thermal conductivity detector. The resulting water was condensed at –60 °C before the gas entered the detector.

The morphology of the surface layer of the samples was studied on the scanning electron microscope Tescan Solaris S900 (Fib/SEM) at a probe electron energy of 2 keV. The microscope was equipped with X-ray energy-dispersive spectrometers ULTIM MAX 100 (Oxford Instruments) with an Aztec analysis system, which provided the qualitative and quantitative analysis. The microanalysis of the samples was carried out at a probe electron energy of 20 keV. Upon deposition of the sample, sputtering of a protective or current-conducting layer was not performed (native conditions) because this could lead to various artefacts in the image at a high magnification. To examine the sample, its particles were fixed on current-conducting carbon adhesive tape.

The fractional composition of gibbsite powder particles, both the initial and after thermal activation and milling in a ball mill, was determined using laser diffraction. The measurements were carried out on a SALD 2101 (Shimadzu Corp., Kyoto, Japan) instrument (the measurement range of 0.03–1000 μm) equipped with a capacity cell. Before being placed in a measuring cell, the preliminarily prepared suspension of the sample in a dispersion medium was subjected to ultrasonication for 2 min and vigorous stirring on a magnetic stirrer for 5 min. The measurement data were processed using Wing-1 software. The employed technique complies with ASTM D4464-15 (Standard Test Method for Particle Size Distribution of Catalytic Materials by Laser Light Scattering).

## 3. Results and Discussion

### 3.1. Fractional Composition of Powder Particles

Figure 1 displays the particle size distribution for the initial gibbsite and after its thermal activation and milling in a ball mill for 6 h. According to the laser diffraction method, the mean particle size of the powder was 100 μm for the initial gibbsite and ca. 15 μm for CTA-G.

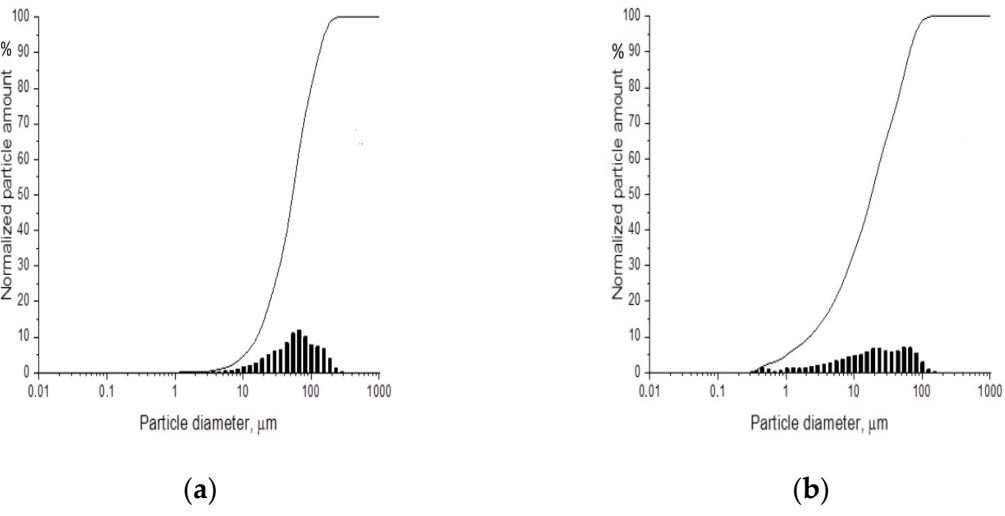

(**a**)  (**b**)

**Figure 1.** Fractional composition of powder particles for the initial gibbsite (**a**) and thermally activated and ball milled for 6 h (**b**).

### 3.2. XRD Data

An earlier XRD study revealed that the initial gibbsite employed for thermoactivation was well crystallized [28–30] and the presence of any other phase admixtures was not detected. As a result of thermoactivation, a diffuse "halo" appears in the diffraction pattern of CTA-G in the angular region of $2\theta = 20$–$40°$, which illustrates the presence of the X-ray amorphous alumina component; one can see the reflections from crystal boehmite (γ-AlOOH) which is formed during centrifugal thermoactivation in the bulk micron-size 3D-single crystals of gibbsite. Peaks of the incompletely decomposed initial gibbsite are also observed [28].

The chemical analysis showed that the nickel content in all the samples was $33 \pm 3$ wt.% (in terms of nickel), which corresponds to the calculated concentration for the synthesis and to the stoichiometric amount of nickel in the system of the "normal" composition $NiAl_2O_4$. Note that in the measurements of such concentrations, the chemical analysis technique gives an error of $\pm 3$ wt.%.

The line at $2\theta = 10°$, indicating the formation of a layered double hydroxide (LDH [19]), appears in diffraction patterns of the dried products of NiAl(25)-110 and NiAl(150)-110 interaction (Figure 2a, curves 1–3). In addition, reflections from the nickel phases, particularly basic nickel nitrates with the composition $Ni_3(NO_3)_2(OH)_x$—BN [18], are superimposed on the overall diffraction pattern; one can see the boehmite (B) and gibbsite phases, which initially entered the composition of the initial CTA-G as residual impurities [28,30]. Since the NiAl(25)-110 and NiAl(150)-110 systems are in the poorly crystallized state, to ensure more accurate attribution of reflections to a certain phase, one of the samples, namely NiAl(150)-110, was washed to remove the occluded admixtures and then dried at 110 °C (Figure 2a, curve 3). Indeed, the LDH phase is observed in the diffraction pattern of the washed sample along with the residual impurities of gibbsite and boehmite (Figure 2a, curve 3). Thus, the washing removes basic nickel nitrate. Note that phase transformations of gibbsite and boehmite residing in CTA-G do not occur under the hydrochemical treatment at room temperature or hydrothermal treatment at 150 °C. This is known from the literature (for example, [25,26]) and was demonstrated earlier in our studies [28,29] on the

hydrothermal treatment at 150 °C of suspensions of the initial gibbsite in a magnesium nitrate solution.

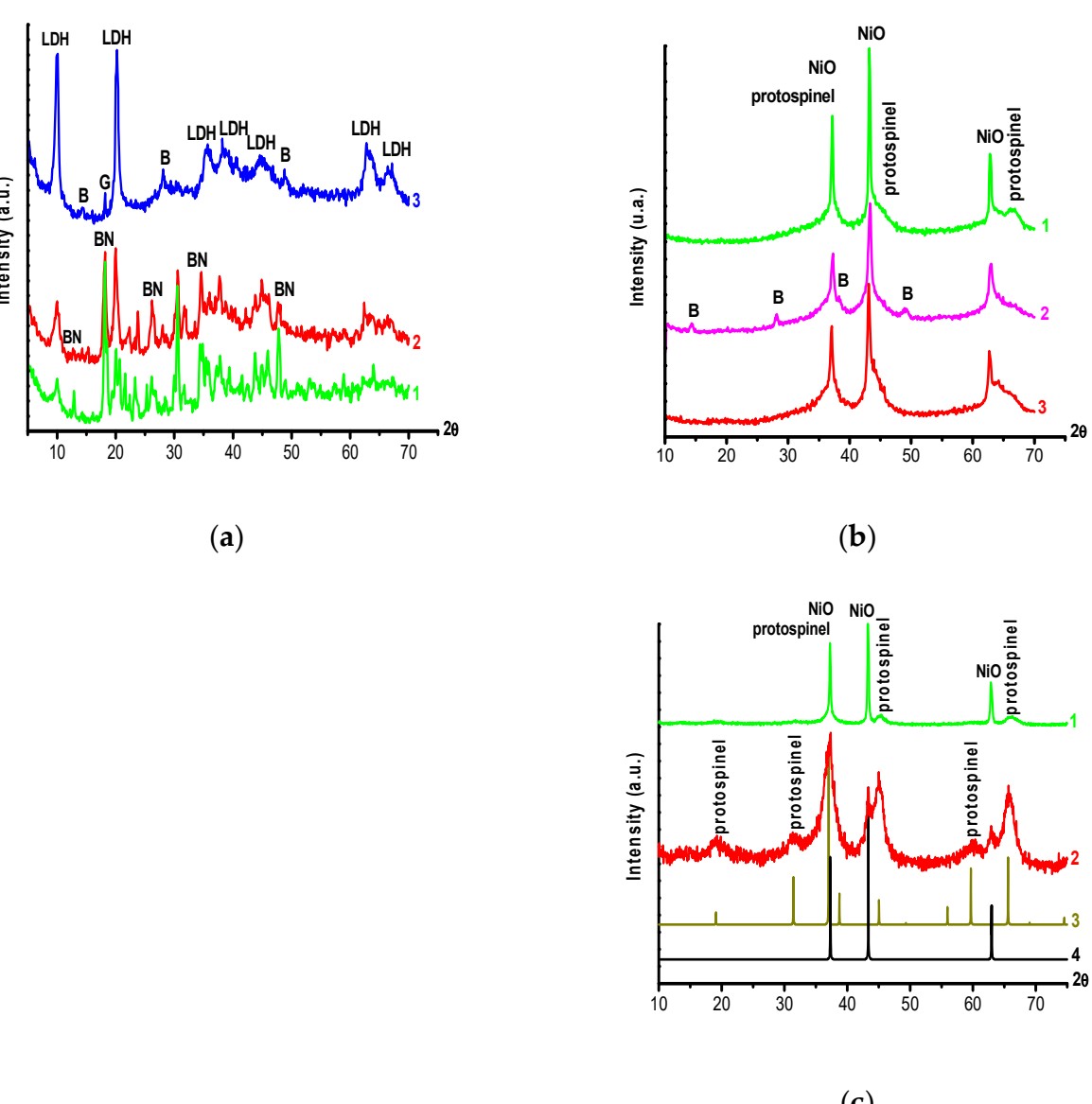

**(a)**

**(b)**

**(c)**

**Figure 2.** Powder diffraction patterns of the samples after low-temperature drying at 110 °C and thermal treatment in the range of 350–850 °C. (**a**)—NiAl(25)-110 (1), NiAl(150)-110 (2), NiAl(150)-110 washed before drying (3); (**b**)—NiAl(25)-550 (1), NiAl(150)-350 (2), NiAl(150)-550 (3); (**c**)—NiAl(25)-850 (1), NiAl(150)-850 (2), the reference sample for well crystallized $NiAl_2O_4$ (3) and the reference sample for well crystallized NiO (4).

According to XRD data, after thermal treatment at 550 °C, only the peaks from NiO phases and $NiAl_2O_4$ "protospinels"—the highly dispersed solid solutions of nickel based on the spinel structure of the low-temperature $\gamma$-$Al_2O_3$—are qualitatively observed in NiAl(25)-550 and NiAl(150)-550 samples (Figure 2b, curves 1–3). Note that in the literature, a "protospinel" implies the compounds in which the molar content of metals, with respect to aluminum, is lower or higher than the stoichiometric one [16,17,19,21,24]. The NiAl(150)-350 sample (Figure 2b, curve 3), calcined at a lower temperature (350 °C), is similar to NiAl(25)-550 and NiAl(150)-550 systems but contains residual boehmite impurities because boehmite completely decomposes at 500–550 °C and, accordingly, is not detected in NiAl(25)-550 and NiAl(150)-550 samples (Figure 2b, curves 1 and 2). After thermal

treatment at 850 °C, two phases, NiO and $NiAl_2O_4$, are also present in NiAl(25)-850 and NiAl(150)-850 samples. The NiO phase is well crystallized in the NiAl(25)-850 sample (Figure 2c, curve 1) and not very crystallized in NiAl(150)-850 (Figure 2c, curve 2). In both cases, $NiAl_2O_4$ "protospinels" are weakly crystallized, which is indicated by the broadened reflections from $NiAl_2O_4$ "protospinels", while more intense reflections from the NiO phase in NiAl(25)-850 qualitatively indicate a higher NiO content in comparison with the hydrothermal sample NiAl(150)-850. The reference samples of individual phases taken from the international X-ray diffraction database ICSD, $NiAl_2O_4$ (ICSD, No. 04_241420, Figure 2c, curve 3) and NiO (ICSD, No. 241420, Figure 2c, curve 4) are well crystallized in comparison with the samples synthesized in our study. The lattice parameter "a" for the well crystallized reference samples NiO and $NiAl_2O_4$, according to the database, is equal to 4.414 and 8.046 Å, respectively. For NiAl(25)-850 and NiAl(150)-850 samples, the calculated lattice parameters "a" have the following values: for NiO and $NiAl_2O_4$ in the composition of NiAl(25)-850—4.178 and 7.994 Å; for NiO and $NiAl_2O_4$ in NiAl(150)-850—4.184 and 8.038 Å, respectively. As follows from the data presented above, the hydrothermal treatment leads to a deeper interaction of nickel with aluminum, which is illustrated by the lattice parameter close to the tabular value for $NiAl_2O_4$ in the NiAl(150)-850 system. On the contrary, for $NiAl_2O_4$ in the NiAl(25)-850 system, the lattice parameter value is lower for both NiAl(150)-850 and for the well crystallized $NiAl_2O_4$ (ICSD, No. 04_241420, Figure 2c, curve 3) which indicates a weaker binding of nickel to the alumina matrix in the NiAl(25)-850 sample synthesized by hydrochemical treatment at room temperature and "normal" pressure. These differences are also considered below and are confirmed by other physicochemical methods. In addition, the sizes of coherent scattering regions (CSR) were estimated from XRD data for NiAl(25)-850 (Figure 2c, curve 1) and NiAl(150)-850 (Figure 2c, curve 2) using characteristic lines of NiO and $NiAl_2O_4$ spinels. The estimated CSR sizes were equal to 1000 Å for NiO and 100 Å for $NiAl_2O_4$ in the composition of NiAl(25)-850; for NiO, the CSR size could not be estimated numerically due to poorly resolved, low-intensity reflections from NiO and $NiAl_2O_4$ at 100 Å in the composition of NiAl(150)-850.

Note that XRD did not detect individual phases of aluminum oxides in the systems calcined at 550–850 °C.

### 3.3. Thermal Analysis Data

According to the thermal analysis data, the initial gibbsite is characterized by a total weight loss of ca. 34 wt.% and the presence of the following endothermal effects: the endo-effect at 97 °C caused by the removal of weakly bound molecular water; the double pre-effect with the maxima at 238 and 256 °C, indicating the initial step of decomposition of plate-like gibbsite crystallites with the formation of boehmite; the endothermal effect of the gibbsite phase dehydration (the peak at 314 °C) and the endo-effect of the boehmite phase decomposition at 536 °C [28,30].

The heating curve of the centrifugal thermal activation product of gibbsite CTA-G shows an endothermal effect with the minimum at 124 °C, which is caused by the removal of molecular water from the X-ray amorphous alumina component of the thermoactivation product. The endo-effect with the minimum at 257 °C, which is caused by dehydration of the residual gibbsite phase, and the endothermal effect with the minimum at 483 °C, which is associated with the decomposition of residual boehmite in $\gamma$-$Al_2O_3$ residing in CTA-G, are preserved. The presence of the exothermal effect with the maximum at 832 °C illustrates the crystallization of the X-ray amorphous phase into the low-temperature forms of $\gamma$-$Al_2O_3$ [25,26,30]. Note that the appearance of exothermal effects with the maxima at 800–850 °C in thermoactivated products, particularly those obtained by centrifugal thermal activation, shows that the X-ray amorphous phase crystallizes into the low-temperature form of $\gamma$-$Al_2O_3$ or $\eta$-$Al_2O_3$ [25,26]. According to the thermal analysis data, CTA-G may have the following composition: G—11 wt.%, boehmite—12 wt.% and X-ray amorphous alumina component—77 wt.%.

Figure 3 displays the thermal analysis data for NiAl(25)-110 and NiAl(150)-110 samples dried at 110 °C to the xerogel state. The samples are characterized by endothermal effects in the region of 100–200 °C, which are related to the removal of weakly bound forms of water in the molecular state of OH-groups. The endothermal effect with the minimum at 285–300 °C is responsible for the removal of nitrates and hydroxyl groups from the LDH structure and the basic nickel nitrate identified above, according to XRD data. This thermal effect also includes weight losses from the residual gibbsite phase. The endo-effect with the extremum at 350–360 °C indicates the completion of thermo-decomposition processes in these systems. In the region from 350–360 °C up to 850 °C, small weight losses appear on the curves in the range of 1.5–4.2 wt.%, which are responsible for the removal of residual hydroxide groups and/or the overall decomposition of boehmite (Figure 3).

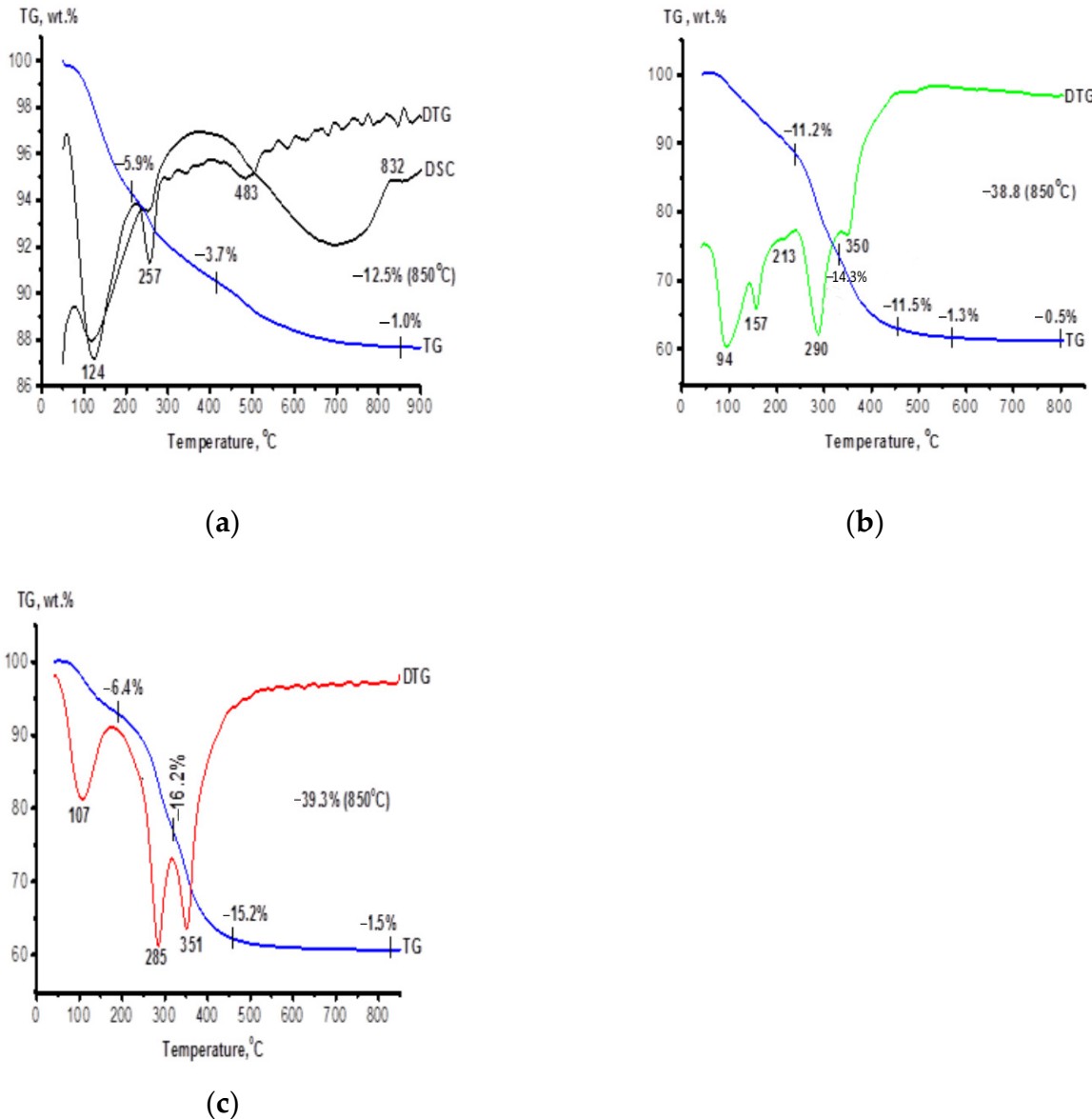

**Figure 3.** Thermal analysis data. (**a**)—the initial CTA-G; (**b**)—NiAl(25)-110; (**c**)—NiAl(150)-110.

A comparison of our earlier results on the interaction of CTA-G with aqueous solutions of magnesium or barium nitrate salts, obtained at the molar cationic ratio of Ba or Mg to Al equal to 1:2 [28–30], with the data acquired in the present study shows that in the case of hydrochemical treatment of CTA-G in aqueous solutions of nickel nitrate, both at room temperature and under hydrothermal conditions at 150 °C, phases of aluminum

hydroxides with pseudoboehmite structures were not formed. If pseudoboehmites were formed in the NiAl(25)-110 and NiAl(150)-110 products, the broadened reflections would be observed in the angular region of 2θ = 14, 29, 38 and 47° on the diffraction patterns, and endothermal effects with the minima would appear in the temperature region of 380–450 °C on the thermal analysis curves [26]. Evidently, the formation of pseudoboehmites should be expected under such synthesis conditions when the nickel content is lower than the stoichiometric one because the interaction of individual CTA-G products in aqueous media, particularly under the common hydrothermal treatment, is known to ensure the formation of pseudoboehmites, bayerites or their mixtures in dependence on the pH of the aqueous medium used for the treatment of CTA-G powders [26].

Table 2 summarizes the qualitative data of thermal analysis and XRD for aluminum–nickel samples NiAl(25)-110 and NiAl(150)-110 after room temperature and hydrothermal interaction in the presence of $Ni^{2+}$ in the solutions with subsequent drying and thermal treatment at 350–850 °C.

**Table 2.** Phase composition of NiAl samples.

| Sample | Qualitative Composition of Samples (XRD + TA) |
|---|---|
| NiAl(25)-110<br>NiAl(150)-110 | LDH, BN, admixture of G and B |
| NiAl(25)-550<br>NiAl(25)-850 | NiO, NiAl$_2$O$_4$ "protospinel" |
| NiAl(150)-350 | NiO, NiAl$_2$O$_4$ "protospinel", B admixture |
| NiAl(150)-550<br>NiAl(150)-850 | NiO, NiAl$_2$O$_4$ "protospinel" |

*3.4. Nitrogen Porosimetry Data*

Figure 4 and Table 3 show the textural characteristics obtained by nitrogen porosimetry. One can see from the desorption curves that NiAl(25)-550 and NiAl(150)-550 samples, after thermal treatment at 550 °C, are characterized by a broad pore size distribution from 3 to 200 nm. An increase in the hydrochemical treatment temperature from room values to 150 °C leads to a growth in the specific surface area (according to BET) up to 141 and 200 m$^2$/g for NiAl(25)-550 and NiAl(150)-550, respectively (Table 3). Therefore, the mean pore diameter remains approximately constant, and the maximum pore volume (0.35 cm$^3$/g) is observed for the NiAl(150)-550 sample obtained under hydrothermal conditions at 150 °C. One can see from the adsorption/desorption isotherms for NiAl(25)-550 and NiAl(150)-550 samples that the absence of a plateau at p/p$_0$~1 indicates the presence of open-ended pores between the particles. This corresponds to hysteresis type H3 according to the IUPAC classification.

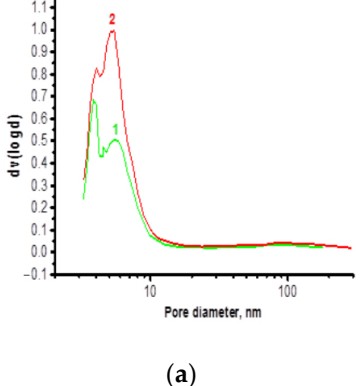

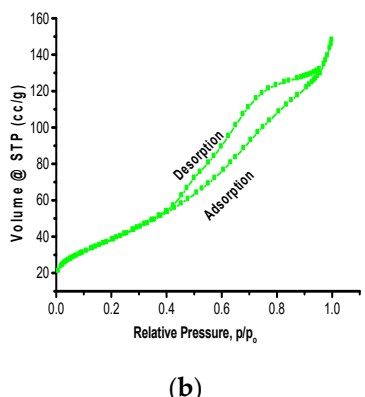

(a)          (b)

**Figure 4.** *Cont.*

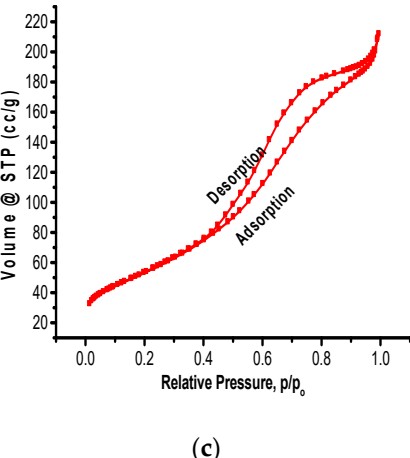

(**c**)

**Figure 4.** Desorption curves of the pore size distribution and adsorption/desorption isotherms. (**a**)—desorption curves of the pore size distribution for NiAl(25)-550 (1) and NiAl(150)-550 (2); (**b**)—adsorption/desorption isotherms for NiAl(25)-550; (**c**)—adsorption/desorption isotherms for NiAl(150)-550.

**Table 3.** Nitrogen porosimetry data.

| Sample | $S_{sp}$, m$^2$/g BET/BJH | $V_{pore}$, cm$^3$/g (BJH) | $D_{pore}$, nm (BJH) |
|---|---|---|---|
| NiAl(25)-550 | 141/162 | 0.24 | 4.0 |
| NiAl(150)-550 | 200/242 | 0.35 | 5.7 |

*3.5. Scanning Electron Microscopy Data*

　　Chemical analysis (mapping) of the surface layer of NiAl(25)-550 and NiAl(150)-550 samples revealed the presence of aluminum and nickel, which was uniformly distributed over the alumina matrix in both variants of the synthesis (Figure 5a,b). According to the scanning electron microscopy data, NiAl(25)-550 and NiAl(150)-550 samples consist of faceted 3D, cubic NiO particles with diameters in the range of 60–400 nm (Figure 6b,d and Figure 7b,d) which reside on the surface of aluminum–nickel "protospinels" identified above by XRD (Figure 2b,c). In the electron microscopy images of the NiAl(25)-550 sample in comparison with NiAl(150)-550, one can see a greater amount of the formed 3D, cubic NiO particles on the alumina support surface, which is also consistent with the X-ray diffraction analysis of NiAl(25)-550 (Figure 2b, curve 1) and NiAl(25)-850 (Figure 2c, curve 1) in which NiO is qualitatively concentrated in a greater amount.

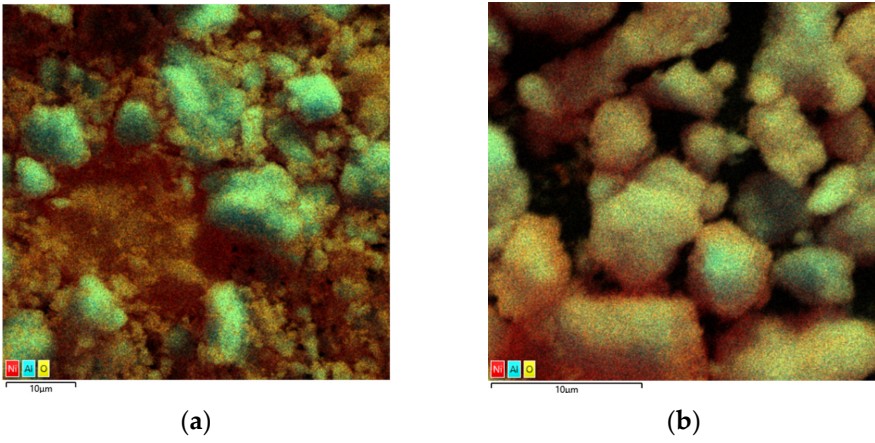

(**a**)　　　　　　　　　　　　　　　　　　　　　(**b**)

**Figure 5.** Chemical analysis of the surface layer of samples. (**a**)—NiAl(25)-550; (**b**)—NiAl(150)-550.

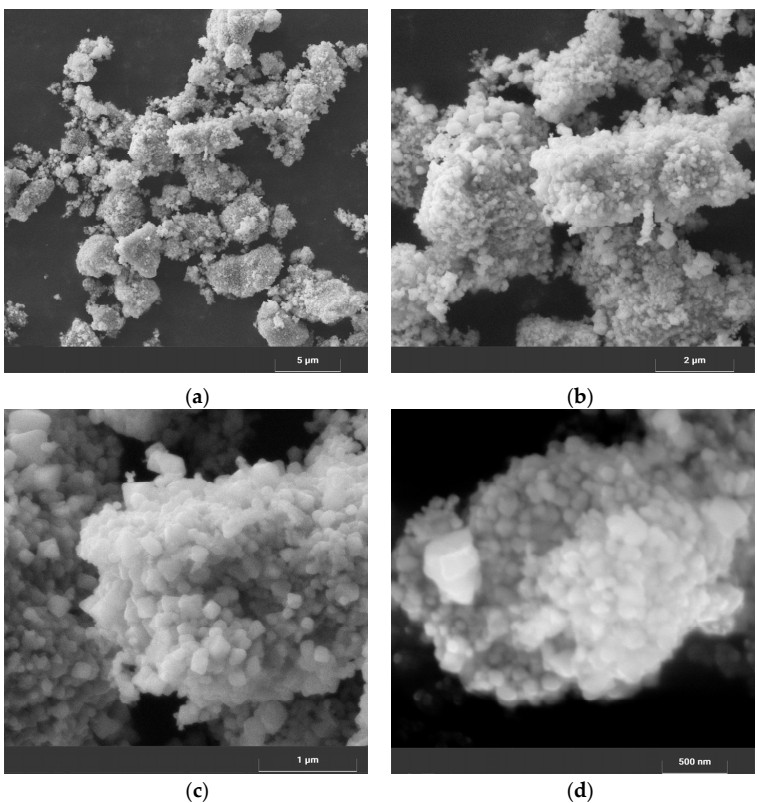

**Figure 6.** Electron microscopy images of the NiAl(25)-550 sample at different magnifications: (**a**)—5 µm; (**b**)—2 µm; (**c**)—1 µm; (**d**)—500 nm.

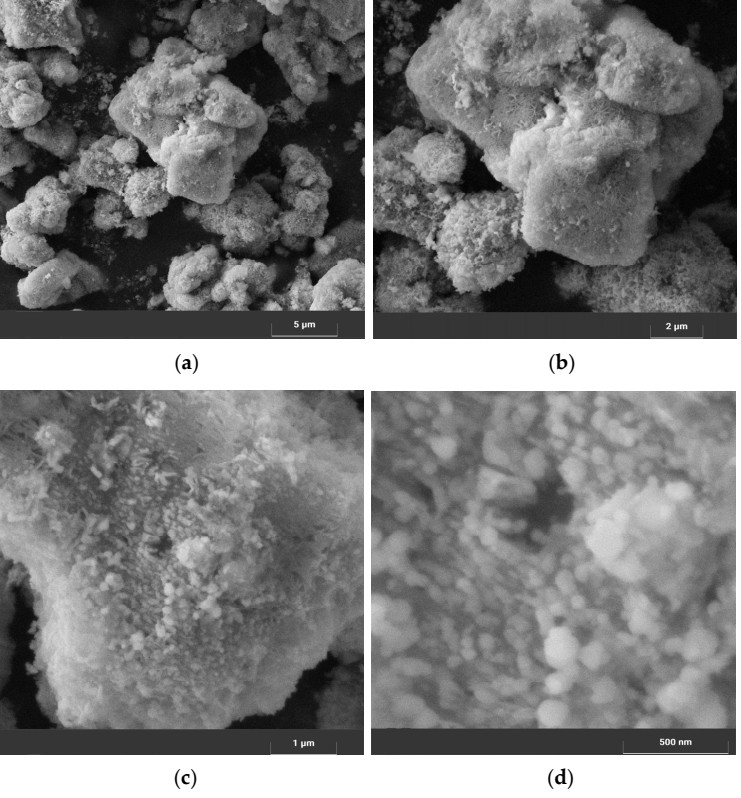

**Figure 7.** Electron microscopy images of the NiAl(150)-550 sample at different magnifications: (**a**)—5 µm; (**b**)—2 µm; (**c**)—1 µm; (**d**)—500 nm.

Thus, it can be concluded from the XRD and scanning electron microscopy data that the main distinction of the room temperature hydrochemical treatment of the nickel solution with CTA-G from the hydrothermal treatment is related to the degree of interaction after thermal treatment. The hydrothermal treatment after calcination ensures virtually compete formation of $NiAl_2O_4$, which may be caused, for example, by the homogenization degree of components during hydrochemical treatments. The room temperature treatment limits the homogenization degree of components by the pore volume (similar to the impregnation step) in contrast to hydrothermal treatment, which ensures a more complete homogenization of the components of the suspensions. The observed differences in the phase composition of mixed oxides are certainly reflected in their reduction characteristics, which are analyzed below by the TPR-$H_2$ method (Figure 8).

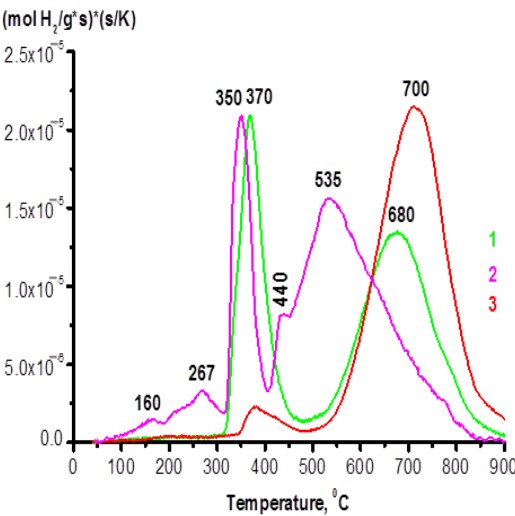

**Figure 8.** TPR-$H_2$ curves. 1—NiAl(25)-550; 2—NiAl(150)-350; 3—NiAl(150)-550.

### 3.6. TPR-$H_2$ Data

In some heterogeneous catalytic processes, aluminum–nickel systems require preliminary activation, the treatment of catalysts in flowing hydrogen, which ensures the formation of active-reduced nickel sites on the surface. Such processes include hydrogenation/dehydrogenation of hydrocarbons [1–4] and the cryogenic reorientation of nuclear spins of the isomers of ortho- and parahydrogen, the so-called ortho–para conversion reaction [10–12]. Thus, the activity of aluminum–nickel systems without activation in hydrogen is much lower compared to those with preliminary activation. It is known that some parameters of the synthesis underlying the formation of the aluminum–nickel system exert a substantial effect on the reduction characteristics of these materials [31–35]. In our study, a relationship between the conditions of the synthesis and nickel reducibility has been established.

Figure 8 displays data of the temperature programmed reduction with hydrogen for NiAl(25)-550, NiAl(150)-350 and NiAl(150)-550 samples. For NiAl(25)-550, we detected two hydrogen absorption peaks in the low-temperature region, with the maximum at 370 °C, and a high-temperature peak at 680 °C (Figure 8, curve 1). The TPR-$H_2$ curve for the NiAl(150)-350 sample (Figure 8, curve 2) has a more complicated profile, which indicates both the presence of nickel oxide particles with different dispersion in the sample and the interaction of nickel with aluminum oxide. In the low-temperature region (below 300 °C), low-intensity peaks are observed at 160 and 273 °C; in the temperature range of 300–900 °C, one can see a narrow, intense peak with the maximum at 350 °C and a broad intense peak with the maximum at 536 °C with a shoulder at 440 °C (Figure 8, curve 2). The same NiAl(150)-550 sample after thermal treatment at 550 °C is characterized by hydrogen absorption peaks similar to NiAl(25)-550 (Figure 8, curve 1). However,

the intensity of the peak at 370 °C is much lower, and the main fraction of hydrogen absorption corresponds to the high-temperature region with the maximum at 700 °C (Figure 8, curve 3). Peaks at 370 °C are caused by the reduction of NiO phases [31–33] observed by XRD (Figure 2b,c). In the high-temperature region, hydrogen absorption peaks of NiAl(25)-550, NiAl(150)-350 and NiAl(150)-550 in the range of 440–700 °C illustrate the reduction of the nickel species that are more strongly bound to the aluminum matrix; these species are the $NiAl_2O_4$ "protospinels" [34,35], which were also revealed above by XRD (Figure 2b,c). Thus, it is seen from the presented TPR-$H_2$ data that nickel reduction is affected not only by the final temperature of the thermally treated samples, which is known from the literature [17–19,31–35], but also by the temperature mode of the treatment of the suspensions.

Table 4 lists such characteristics of the reduction of aluminum–nickel samples as the total integrated areas of hydrogen absorption, the integrated areas of absorption corresponding to individual absorption peaks and the temperature maxima of absorption. One can see from the obtained TPR-$H_2$ data that the maximum amount of absorbed hydrogen corresponds to the NiAl(150)-350 sample (Figure 8, curve 2). Therefore, for the NiAl(150)-550 sample after thermal treatment at 550 °C, the total amount of absorbed hydrogen is approximately 20% lower and the main absorption degree corresponds to the high-temperature region with the maximum at 700 °C. The amount of hydrogen absorbed by the NiAl(25)-550 sample (Figure 8, curve 1) is close to that absorbed by NiAl(150)-550, which was synthesized under hydrothermal conditions. However, the amount of hydrogen absorbed in the low-temperature region (350–370 °C) is much higher and close to that obtained for NiAl(150)-350 (Figure 8, curve 2, Table 4).

**Table 4.** Characteristics of the reduction of aluminum–nickel samples according to TPR-$H_2$ data.

| Sample | $T_{max}$, °C | $H_2$ Absorption with Respect to NiO, % | $H_2$ Absorption with Respect to $NiAlO_x$, % | Amount of Absorbed $H_2$ per g NiO, $10^{-3}$ mol/g |
|---|---|---|---|---|
| NiAl(150)-350 | 160–270 | 8 | - | 0.38 |
| | 350 | 22 | - | 1.11 |
| | 535 | - | 70 | 3.43 |
| | | | | $\sum = 4.92$ |
| NiAl(25)-550 | 370 | 36 | - | 1.40 |
| | 680 | - | 64 | 2.46 |
| | | | | $\sum = 3.86$ |
| NiAl(150)-550 | 370 | 5 | - | 0.20 |
| | 700 | - | 95 | 3.78 |
| | | | | $\sum = 3.98$ |

Figure 9 shows a scheme reflecting the main technological steps in the synthesis of aluminum–nickel systems using the classical coprecipitation method and the centrifugally thermoactivated CTA-G product. One can see that the synthesis of highly loaded Ni-aluminum materials with the stoichiometric composition and the total nickel content of ca. 33 wt.% based on CTA-G ensures a decrease in the number of synthesis steps and prevents the formation of wastewater.

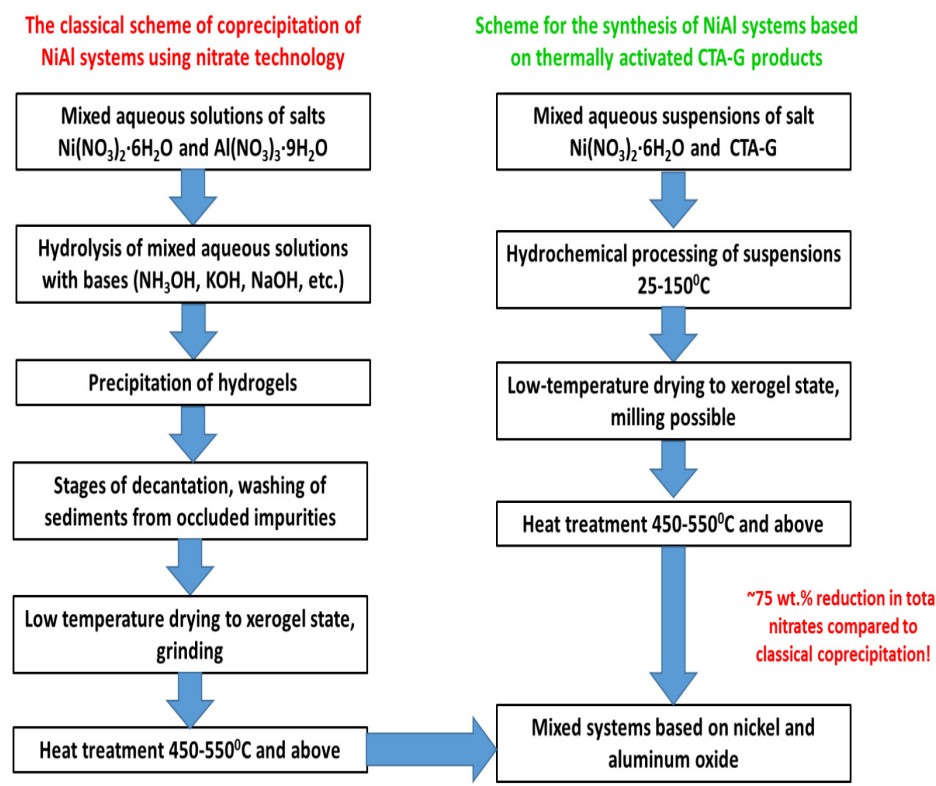

**Figure 9.** A flow chart of the main technological steps in the synthesis of complex NiAl systems using a classical coprecipitation method and a comparison with the use of CTA-G product.

## 4. Conclusions

Using the X-ray, thermal, microscopy, adsorption and chemical analysis methods, we have demonstrated that highly loaded, mixed aluminum–nickel oxide systems with a nickel content of ca. 33 wt.% can be synthesized by the hydrochemical treatment at room temperature or by the hydrothermal treatment at 150 °C of suspensions of the product of centrifugal thermal activation of gibbsite in aqueous solutions of nickel nitrate with subsequent drying and thermal treatment of the produced xerogels. It was shown that thermal treatment of the hydrothermal interaction products in the range of 350–850 °C leads to the formation of NiO phases and highly dispersed solid solutions of nickel based on the $NiAl_2O_4$ spinel structure with a high specific surface area of 140–200 $m^2/g$. The ratio of phases and their dispersion are determined by the treatment conditions of suspensions and by the final calcination temperature of the formed xerogels. The room temperature hydrochemical treatment ensures the predominant formation of free NiO (ca. 35% according to the TPR-$H_2$ data) on the "protospinel" surface, whereas the hydrothermal treatment at 150 °C leads to a deeper interaction of components, resulting in the predominant formation of poorly crystallized $NiAl_2O_4$ "protospinel" (90% according to the TPR-$H_2$ data) with a high (up to 200 $m^2/g$) specific surface area (90% according to the TPR data) after thermal treatment at 550 °C.

The results obtained form a solid basis for the synthesis of highly loaded aluminum–nickel catalysts using the centrifugally thermoactivated products of gibbsite, particularly with the stoichiometric composition, by the low-waste method, ensuring a decrease in the amount of initial reagents, the number of synthesis steps, the amount of wastes and the total amount of nitrates compared to the classical nitrate scheme of coprecipitation of these compounds with the similar phase and chemical composition.

**Author Contributions:** Conceptualization, methodology, writing—original and final draft preparation, A.V.Z.; data curation, L.A.I.; SEM investigation, E.A.S.; XRD investigation, A.S.G. All authors have read and agreed to the published version of the manuscript.

**Funding:** The study was financially supported by the Russian Science Foundation within project No. 23-23-00241.

**Data Availability Statement:** Not applicable.

**Acknowledgments:** The authors acknowledge core facilities "VTAN" (Novosibirsk State Univerity) for access to its experimental equipment.

**Conflicts of Interest:** There is no conflict of interest in the submission of this manuscript.

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
