# Peer review of "Low-Waste Synthesis and Properties of Highly Dispersed NiO·Al2O3 Mixed Oxides Based on the Products of Centrifugal Thermal Activation of Gibbsite"

_2305-7084, doi:10.3390/chemengineering7040071_

Round 1

Reviewer 1 Report

Zhuzhgov et al. presented their findings on the thermos treatment of xerogels at various temperatures, resulting in the formation of NiO and NiAl2O3 with different ratios and surface areas. However, the manuscript exhibits many careless issues, such as inconsistent line spacing, no reference in line 194 (even highlighted by authors), etc. I find this lack of meticulousness and seriousness concerning regarding the authors' work. Consequently, I have reservations about the validity of the results presented in the manuscript, and I do not believe it is suitable for acceptance

Comment 1: The particle size of NiO and Al2O3 should be provided with a detailed calculation method.

Comment 2: It is recommended to conduct a detailed exploration of the porosity of the solid solution. This can be achieved by providing N2 isotherms and discussing the types of isotherms/hysteresis loops, which will offer valuable insights into the morphological properties of the materials. Additionally, it would be crucial to elucidate the relationship between different temperature treatments and the resulting morphologies, as this would contribute to understanding the mechanism of solid solution formation.

Comment 3: Appropriate references should be cited in the discussion section. For instance, Line 194 lacks a reference for the XRD peak positions, and when discussing the H2-TPR results, references should be provided for the interpretation of reduction peaks.

Extensive editing of English language required

Author Response

Dear Reviewer 1,

We would like to thank you for the careful and insightful review of the manuscript. Please find our responses to each of your comments below.

1. Comment 1: Zhuzhgov et al. presented their findings on the thermos treatment of xerogels at various temperatures, resulting in the formation of NiO and NiAl2O3 with different ratios and surface areas. However, the manuscript exhibits many careless issues, such as inconsistent line spacing, no reference in line 194 (even highlighted by authors), etc. I find this lack of meticulousness and seriousness concerning regarding the authors' work. Consequently, I have reservations about the validity of the results presented in the manuscript, and I do not believe it is suitable for acceptance.

Response 1: Links have been added on line 194 (page 6) and line spacing adjusted to journal requirements. In addition, the numbering of tables, figures and references to the literature was checked according to the text of the article. All changes are highlighted yellow.

2. Comment 2: The particle size of NiO and Al2O3 should be provided with a detailed calculation method.

Response 2: The fractional composition of gibbsite powder particles, both the initial and after thermal activation and milling in a ball mill, was determined using laser diffraction. The measurements were carried out on a SALD 2101 (Shimadzu Corp., Japan) instrument (the measurement range of 0.03-1000 µm) equipped with a capacity cell. Before placing in a measuring cell, the preliminarily prepared suspension of the sample in a dispersion medium was subjected to ultrasonication for 2 minutes and vigorous stirring on a magnetic stirrer for 5 minutes. The measurement data were processed using the Wing-1 software. The employed technique complies with ASTM D4464-15 (Standard Test Method for Particle Size Distribution of Catalytic Materials by Laser Light Scattering). Figure 1 displays the particle size distribution for the initial gibbsite and after its thermal activation and milling in a ball mil for 6 hours. According to the laser diffraction method, the mean particle size of the powder was 100 µm for the initial gibbsite, and ca. 15 µm for СTA-G.  The changes are highlighted in yellow on page 5.

In addition, sizes of coherent scattering regions (CSR) were estimated from XRD data for NiAl(25)-850 (Figure 2c, curve 1) and NiAl(150)-850 (Figure 2c, curve 2) using characteristic lines of NiO and NiAl2O4 spinels. The estimated CSR sizes were equal to 1000 Å for NiO and 100 Å for NiAl2O4 in the composition of NiAl(25)-850; for NiO, the CSR size could not be estimated numerically due to poorly resolved low-intensity reflections from NiO and NiAl2O4 at 100 Å in the composition of NiAl(150)-850. The changes are highlighted in yellow on page 5.

According to the scanning electron microscopy data, NiAl(25)-550 and NiAl(150)-550 samples consist of faceted 3D-cubic NiO particles with the diameter in the range of 60-400 nm (Figures 6b, d and 7b, d), which reside on the surface of aluminum-nickel “protospinels” identified above by XRD (Figure 2b, c). The size range of NiO is determined based on the scale bar on SEM images. In the scanning electron microscopy images of the NiAl(25)-550 sample (Figures 6b, d) in comparison with NiAl(150)-550 (Figures 7b, d), one can see a greater amount of the formed 3D-cubic NiO particles on the alumina support surface, which is also consistent with the X-ray diffraction analysis of NiAl(25)-550 (Figure 2b, curve 1) and NiAl(25)-850 (Figure 2c, curve 1), in which NiO is qualitatively concentrated in a greater amount, compared to hydrothermal sample NiAl(150)-550 (Figure 2b, curve 3) and NiAl(150)-850 (Figure 2c, curve 2).

3. Comment 3: It is recommended to conduct a detailed exploration of the porosity of the solid solution. This can be achieved by providing N2 isotherms and discussing the types of isotherms/hysteresis loops, which will offer valuable insights into the morphological properties of the materials. Additionally, it would be crucial to elucidate the relationship between different temperature treatments and the resulting morphologies, as this would contribute to understanding the mechanism of solid solution formation.

Response 3: In Figure 4 (page 10), we added desorption/absorption isotherms of the same NiAl(25)-550 and NiAl(150)-550 samples to the desorption curves of pore size distribution. One can see from the adsorption/desorption isotherms for NiAl(25)-550 and NiAl(150)-550 samples that the absence of a plateau at p/p~ 1 indicates the presence of open-ended pores between the particles. This corresponds to hysteresis type H3 according to the IUPAC classification.

4. Comment 4: Appropriate references should be cited in the discussion section. For instance, Line 194 lacks a reference for the XRD peak positions, and when discussing the H2-TPR results, references should be provided for the interpretation of reduction peaks.

Response 4: When discussing the results of XRD and H2-TPR, links are provided for the XRD peak position and interpretation of reduction peaks. The changes are highlighted yellow on page 6 (line 194) and pages 13, 14.

Yours sincerely,

A. Zhuzhgov on behalf of the co-authors

Reviewer 2 Report

1. It seems to me that these samples, characterized only by X-ray diffraction, are still insufficiently characterized. It would be nice to see data on Raman, infrared and even on photoluminescence in order to be sure that there are no other phases there or they will be visible on Raman or on luminescence. In this form, the article is based more on the wishes of the authors of what they believe ...

2. A lot is said about OH, but for this you need to show  IR spectra.

3. Explanation of figure 3 is clearly insufficient. How many types, what dynamics - complete silence.  It is possible that it is necessary to make a detailed analysis of the pores by the PAL method, as was recently demonstrated for MgAl2O4  ( Klym, H.; Karbovnyk, I.; Piskunov, S.; Popov, A.I. Positron Annihilation Lifetime Spectroscopy Insight on Free Volume Conversion of Nanostructured MgAl2O4 Ceramics. Nanomaterials 202111, 3373. https://doi.org/10.3390/nano11123373 )

Author Response

Dear Reviewer 2,

We would like to thank you for the careful and insightful review of the manuscript. Please find our responses to each of your comments below.

1. Comment 1: It seems to me that these samples, characterized only by X-ray diffraction, are still insufficiently characterized. It would be nice to see data on Raman, infrared and even on photoluminescence in order to be sure that there are no other phases there or they will be visible on Raman or on luminescence. In this form, the article is based more on the wishes of the authors of what they believe ...

Response 1: In the present work, the samples were not characterized by IR-spectroscopy and/or Raman. In addition, we note that since the NiAl samples in this work were synthesized with a content of ~33 wt.% Ni, which corresponds to the stoichiometric content in the NiAl2O4 system, no luminescence will be observed for such aluminum systems with a high nickel content. Yes, there are data in the literature on luminescence, in which Ni2+ is used as a structure-sensitive probe in a number of systems, but nickel concentrations in them are extremely low (1-5 wt.% and below) [for example, Brik, M.G. Crystal field analysis of the absorption spectra and electron-phonon interaction in Ca3Sc2Ge3O12:Ni2+. Journal of Physics and Chemistry of Solids 2006, 67, 738-744. https://doi.org/10.1016/j.jpcs.2005.11.005; Krsmanovi´c, R.M., Anti´c, M.M., Drami´canin, M.D., Brik, M.G. Structural, spectroscopic and crystal field analyses of Ni2+ and Co2+ doped Zn2SiO4 powders. Appl Phys A 2011, 104, 483-492. https://doi.org/10.1007/s00339-011-6291-6 .].    

At the same time, from the point of view of the persuasiveness of the presence or absence of other phases besides NiO and NiAl2O4, identified by XRD (Figure 2b, c, page 7), it is quite well possible to operate with TPR-H2 data here (Figure 8, page 14). Peaks at 370°С are caused by the reduction of NiO phases observed by XRD (Figure 2b, с). In the high-temperature region, hydrogen absorption peaks of NiAl(25)-550, NiAl(150)-350 and NiAl(150)-550 in the range of 440-700°С testify to the reduction of the nickel species that are more strongly bound to the aluminum matrix; these species are the NiAl2O4 “protospinels”, which were also revealed above by XRD (Figure 2b, c). Note that XRD and TPR-H2 data did not detect individual phases of aluminum oxides in the systems calcined at 550-850°С.

2. Comment 2: A lot is said about OH, but for this you need to show IR spectra....

Response 2: OH-groups are discussed in the section where thermograms of NiAl(25)-110 and NiAl(150)-110 samples (Figure 3, page 9) before heat treatment (dried at 1100C) are considered. There are a large amount of data in the literature on the interpretation of thermograms of aluminum systems (obtained by different methods) containing transition and non-transition metals, including Ni-aluminum [for example, Kobzar, E.O.; Stepanova, L.N.; Leont’eva, N.N.; Belskaya; O.B. The influence of composition of Ni-containing layered hydroxides prepared by mechanochemical method on their properties in the furfural hydrogenation // AIP Conf. Proceed 2020, 2301. https://doi.org/10.1063/5.0032858; Prakash I., Muralidharan P., Nallamuthu N., Satyanarayana N. Preparation of NiAl2O4/SiO2 and Co2+-doped NiAl2O4/SiO2 nanocomposites by the sol-gel route // J. Am. Ceram. Soc 2006, 89, 2220-2225. https://doi.org/10.1111/j.1551-2916.2006.00993.x; Jabłońska, M; Nothdurft, K; Nocuń, M; Girman, Vl; Palkovits, R. Redox-performance correlations in Ag–Cu–Mg–Al, Ce–Cu–Mg–Al, and Ga–Cu–Mg–Al hydrotalcite derived mixed metal oxides. Applied Catalysis B: Environmental 2017, 207, 385-396. https://doi.org/10.1016/j.apcatb.2017.01.079; Kong, L.B., Li, X.M, Liu, M.C., Luo, Y.C., Kang, L. Fabrication of flower-like Ni3(NO3)2(OH)4 and their electrochemical properties evaluation // Mat. Res. Bull 2012, 47, 1641-1647. https://doi.org/10.1016/j.materresbull.2012.03.051.]. Note that each temperature region is responsible for the removal of different forms of water during thermal heating. The temperature range of 100-2000С refers to the removal of molecular water, the higher temperature range of 250-3000С is due to partial or complete dehydration, and the range of 500-5500С is responsible for the complete thermal transformation (in our case Figure 3, page 9) and the transition of the system to an individual or mixed oxide state (NiO·Al2O3), what is detected by XRD after heat treatment (Figure 2 b, c).

3. Comment 3: Explanation of figure 3 is clearly insufficient. How many types, what dynamics - complete silence. It is possible that it is necessary to make a detailed analysis of the pores by the PAL method, as was recently demonstrated for MgAl2O4  ( Klym, H.; Karbovnyk, I.; Piskunov, S.; Popov, A.I. Positron Annihilation Lifetime Spectroscopy Insight on Free Volume Conversion of Nanostructured MgAl2O4 Nanomaterials 2021, 11, 3373. https://doi.org/10.3390/nano11123373).

Response 3:  We thank the referee for the comment and the link dedicated to the detailed study of MgAl2O4 by PAL method. However, the purpose of this work was not a detailed study of the porous structure, and the method of nitrogen porosimetry was used as an addition to the available physicochemical analysis method to characterize the value of the specific surface area, pore size distribution. At the same time, taking into account the comment of the reviewer, in Figure 4 (page 10) we added desorption/absorption isotherms of the same NiAl(25)-550 and NiAl(150)-550 samples to the desorption curves of pore size distribution. One can see from the adsorption/desorption isotherms for NiAl(25)-550 and NiAl(150)-550 samples that the absence of a plateau at p/p~ 1 indicates the presence of open-ended pores between the particles. This corresponds to hysteresis type H3 according to the IUPAC classification. The changes are highlighted yellow on page 10.

Yours sincerely,

A. Zhuzhgov on behalf of the co-authors

Reviewer 3 Report

This paper reported that the highly loaded mixed Al-Ni oxide systems can be synthesized using two methods, i.e. the hydrochemical treatment at room temperature and hydrothermal treatment of suspensions regarding the aqueous solutions of nickel nitrate. Additionally, the synthesized NiAl oxides were characterized by a series of analysis methods, including X-ray, thermal, microscopy, adsorption and chemical analysis. Overall, the manuscript is well-structured, but there is room for minor improvement in the use of English expressions. The following revision suggestions are provided to enhance the quality of the work.

1.      Introduction: The advantages of the synthesis method applied in this work should be in more detail, compared to other synthesis methods for AlNi materials.

2.       The definitions of all the sample names such as NiAl(25)-110, NiAl(150)-110 and NiAl(150)-110 mentioned in this work should be listed in a table to improve reader comprehension.

3.       Figure 2: What are the main differences between the initial thermal analysis result and the results for Al-Ni samples?

4.       Figure 2: How do the authors determine the different temperature regions containing 100-200 C, 285-300 C and 285-360 C, and infer the possible chemical bonds relevant to each temperature region?

5.      Line 320-323: Is NiAl2O4 the only substance formed in the hydrothermal treatment? What specific differences are caused by the homogenization degrees of the two methods?

There is room for minor improvement in the use of English expressions. 

Author Response

Dear Reviewer 3,

We would like to thank you for the careful and insightful review of the manuscript. Please find our responses to each of your comments below.

1. Comment 1: Introduction: The advantages of the synthesis method applied in this work should be in more detail, compared to other synthesis methods for AlNi materials.

Response 1: In the introduction on pages 2 and 3, the advantages of the AlNi synthesis method used in this work, compared with other methods for the synthesis of AlNi materials, were described in more detail. The сhanges are highlighted in yellow.

2. Comment 2: The definitions of all the sample names such as NiAl(25)-110, NiAl(150)-110 and NiAl(150)-110 mentioned in this work should be listed in a table to improve reader comprehension.

Response 2: Designations of samples used in the study are described below in Table 1. The changes are highlighted yellow on page 4.

3. Comment 3: Figure 2: What are the main differences between the initial thermal analysis result and the results for Al-Ni samples?.

Response 3:  The heating curve of the centrifugal thermal activation product of gibbsite CTA-G shows the endothermal effect with the minimum at 124°С, which is caused by the removal of molecular water from the X-ray amorphous alumina component of the thermoactivation product. The endo-effect with the minimum at 257°С, which is caused by dehydration of the residual gibbsite phase, and the endothermal effect with the minimum at 483°С, which is associated with the decomposition of residual boehmite in γ-Al2O3 residing in CTA-G, are preserved (Figure 3a). The presence of the exothermal effect with the maximum at 832°С testifies to crystallization of the X-ray amorphous phase into the low-temperature forms of γ-Al2O3. Note that the appearance of exothermal effects with the maxima at 800-850°С in thermoactivated products, particularly those obtained by centrifugal thermal activation, shows that the X-ray amorphous phase crystallizes into the low-temperature form of γ-Al2O3 or η-Al2O3. As a result of hydrochemical treatment of CTA-G in solutions of nickel nitrate at room temperature and under hydrothermal conditions, the thermograms of hydration products show significant changes (Figure 3b, c). The disappearance of the exothermic effect at 8320С on the thermograms of NiAl(25)-110 and NiAl(150)-110 indicates crystallization with the formation of new products LDH, BN, as evidenced by the appearance of endothermic effects at 285-3500С (Figure 3b, c) . The XRD data of the NiAl(25)-110 and NiAl(150)-110 samples (Figure 2a) confirm the formation of crystallization products LDH, BN, which is in agreement with the thermal analysis data. Note that gibbsite and boehmite impurities in the NiAl(25)-110 and NiAl(150)-110 hydration products are present according to XRD data (Figure 2a), since they were originally present in CTA-G. Under the conditions of hydrothermal treatment at 1500C, as well as at room temperature, gibbsite and boehmite do not crystallize –  more stringent processing conditions are needed starting from a temperature of  2500C, at which gibbsite can already transform into bemeth (pseudo-boehmite) structures [Panasyuk, G.P.; Belan, V.N.; Voroshilov, I.L.; Kozerozhets, I.V. Hydrargillite → Boehmite transformation. Inorganic Materials 2010, 46, 747-753. https://doi.org/10.1134/S0020168510070113.]. According to XRD, gibbsite residues disappear after 300-3500C (Figure 2b, curve 2), and boehmite at 500-5500C (Figure 2b, curve 1, 3), which is combined with thermal transformations at the heating inlet for NiAl(25)-110 and NiAl(150)-110 according to the thermal analysis of these compounds (Figure 3b, c).

4. Comment 4: Figure 2: How do the authors determine the different temperature regions containing 100-200C, 285-300C and 285-360C, and infer the possible chemical bonds relevant to each temperature region?.

Response 4:  As was discussed in comment 3 above, firstly, the assignment of one or another temperature region on the thermograms of calcined and uncalcined samples is done in conjunction with X-ray phase analysis of the same samples. At second, there is a large amount of data in the literature on the interpretation of thermograms of aluminum systems (obtained by different methods) containing transition and non-transition metals, including Ni-aluminum [for example, Kobzar, E.O.; Stepanova, L.N.; Leont’eva, N.N.; Belskaya; O.B. The influence of composition of Ni-containing layered hydroxides prepared by mechanochemical method on their properties in the furfural hydrogenation // AIP Conf. Proceed 2020, 2301. https://doi.org/10.1063/5.0032858; Prakash I., Muralidharan P., Nallamuthu N., Satyanarayana N. Preparation of NiAl2O4/SiO2 and Co2+-doped NiAl2O4/SiO2 nanocomposites by the sol-gel route // J. Am. Ceram. Soc 2006, 89, 2220-2225. https://doi.org/10.1111/j.1551-2916.2006.00993.x; Jabłońska, M; Nothdurft, K; Nocuń, M; Girman, Vl; Palkovits, R. Redox-performance correlations in Ag–Cu–Mg–Al, Ce–Cu–Mg–Al, and Ga–Cu–Mg–Al hydrotalcite derived mixed metal oxides. Applied Catalysis B: Environmental 2017, 207, 385-396. https://doi.org/10.1016/j.apcatb.2017.01.079; Kong, L.B., Li, X.M, Liu, M.C., Luo, Y.C., Kang, L. Fabrication of flower-like Ni3(NO3)2(OH)4 and their electrochemical properties evaluation // Mat. Res. Bull 2012, 47, 1641-1647. https://doi.org/10.1016/j.materresbull.2012.03.051.]. Note that each temperature region is responsible for the removal of different forms of water during thermal heating. The temperature range of 100-2000С refers to the removal of molecular water, the higher temperature range of 250-3000С is due to partial or complete dehydration, and the range of 500-5500С is responsible for the complete thermal transformation (in our case Figure 3, page 9) and the transition of the system to an individual or mixed oxide state (NiO·Al2O3), what is detected by XRD after heat treatment (Figure 2 b, c).

5. Comment 5 : Line 320-323: Is NiAl2O4 the only substance formed in the hydrothermal treatment? What specific differences are caused by the homogenization degrees of the two methods?

Response 5:  Comparing the XRD data (Figure 2c, curve 1, 2) of the NiAl(25)-8500C and NiAl(150)-8500C calcination products synthesized at room temperature and under hydrothermal treatment conditions, it is qualitatively seen that NiAl(25)-8500C mainly consists of NiO and spinels of the NiAl2O4 type (Figure 2c, curve 1), and a sample of spinel of the NiAl(150)-8500C type with traces of NiO (Figure 2c, curve 2). At room temperature, incomplete crystallization of CTA-G occurs, therefore, as a result, nickel nitrate is deposited on the hydrated Al-CTA-G matrix. During hydrothermal treatment, homogenization is higher, and due to the temperature, a deeper incorporation of nickel into the aluminum matrix during hydrochemical treatment at 1500C is ensured. This can also be seen from the lattice parameters “a”, which is higher for the hydrothermal sample NiAl(150)-8500C (8.038 Å) compared to the room synthesis of NiAl(25)-8500C (7.994 Å). Note that the lattice parameter “a” for the well crystallized reference samples NiO and NiAl2O4, according to the database, is equal to 4.414 and 8.046 Å, respectively. For NiAl(25)-850 and NiAl(150)-850 samples, the calculated lattice parameters “a” have the following values: for NiO and NiAl2O4 in the composition of NiAl(25)-850, 4.178 and 7.994 Å; for NiO and NiAl2O4 in NiAl(150)-850, 4.184 and 8.038 Å, respectively (page 6 in the article). Note that XRD did not detect individual phases of aluminum oxides in the systems calcined at 550-850°С.

Yours sincerely,

A. Zhuzhgov on behalf of the co-authors

Round 2

Reviewer 1 Report

Accept the current version

 Moderate editing of English language required

Reviewer 2 Report

After revision, this manuscript can be recommended for publication.

Reviewer 3 Report

The revised manuscript should be accepted.

The full manuscript needs a minor editing of English language.